# Percutaneous Ultrasound-Guided Carotid Access and Puncture Closure with Angio-Seal in Horses

**DOI:** 10.3390/ani12121481

**Published:** 2022-06-07

**Authors:** Arantza Vitoria, Alicia Laborda, Carolina Serrano-Casorrán, Sara Fuente, Antonio Romero, Francisco José Vázquez

**Affiliations:** 1Veterinary Hospital of the University of Zaragoza (HVUZ), C/Miguel Servet 177, 50013 Zaragoza, Spain; avm@unizar.es (A.V.); alaborda@unizar.es (A.L.); carolse@unizar.es (C.S.-C.); sfuente@unizar.es (S.F.); aromerol@unizar.es (A.R.); 2Department of Animal Pathology, Faculty of Veterinary, University of Zaragoza, 50009 Zaragoza, Spain

**Keywords:** interventional radiology, endovascular interventions, arterial access, ultrasound-guided, closure device, Angio-Seal, horses

## Abstract

**Simple Summary:**

Endovascular surgery is a type of image-guided minimally invasive surgery that aims to solve different types of pathologies from inside the blood vessel. This technique requires access to a peripheral vessel (vein or artery) and the navigation through the vascular system to reach the operating site, using different catheters. In horses, the main indication for endovascular therapy involves access to the common carotid artery for different purposes. Traditionally, this was performed by open dissection of the neck to reach and incise the common carotid artery, followed by vascular suture of the vessel and skin closure when the procedure was over. However, this can also be performed by percutaneous puncture of the artery, using ultrasound to guide the needle to its adequate position in the artery, as has been reported in experimental horses. Along with that, in human medicine, commercial closure systems are used to seal these arterial punctures and avoid some complications (mainly haematomas and haemorrhages). We describe our experience in a series of 11 clinical cases in which this minimally invasive way of access was used, and the puncture site was sealed with one of these devices (AngioSeal arterial closure system), reporting its first use in horses. In all cases, the artery was effectively accessed, and the planned procedure could be performed. Our haematoma/bleeding rate (16.66%) was lower than in other studies using the same type of access, even considering two failures due to incorrect use of the device. However, further studies comparing AngioSeal use to simple manual compression would be necessary to be able to recommend their routine use in horses.

**Abstract:**

**Background:** There are different indications for endovascular surgery in horses, mainly the treatment of guttural pouch mycosis. Traditionally, these procedures are carried out by open arteriotomy of the common carotid artery (CCA), although less invasive percutaneous ultrasound-guided carotid access (PUGCA) has been described in experimental horses. In human medicine, commercial closure systems are used to seal these arterial puncture sites and reduce complications. The aims of this study are to retrospectively describe our experience with PUGCA in clinical cases and to report, for the first time, the use of the commercial vascular closure device Angio-Seal after PUGCA in horses. **Methods:** Retrospective study of clinical case records. Collected parameters, including the feasibility of the PUGCA and variables related to the safety and efficacy of the use of the Angio-Seal. **Results:** Twelve PUGCA procedures in 11 horses were included. In all cases, the artery was effectively accessed, and the planned procedure could be performed. In two cases, haematoma/bleeding due to incorrect use of the Angio-Seal was recorded. This complication rate (16.66%) was lower than that obtained in other studies using PUGCA in horses, but where the puncture was sealed by manual compression only. **Main limitations:** A control group of clinical cases with PUGCA but without using Angio-Seal is not available. **Conclusions:** Clinical data confirm previous experimental results, which showed that PUGCA is safe and effective in horses. The Angio-Seal system, regardless of possible complications due to incorrect use, can be used safely and effectively in horses. Further studies comparing arterial access site management using manual compression or Angio-Seal would be necessary to state if its routine use in horses is advisable.

## 1. Introduction

Percutaneous access to medium or large arterial vessels (femoral, brachial and carotid arteries) by the means of introducer sheaths is more and more common in animal experimentation and in veterinary practice [1]. In horses, the embolisation of the vessels surrounding the guttural pouch wall for the endovascular treatment of guttural pouch mycosis is the most common procedure using this arterial access [2,3]. Other indications for the use of these accesses, such as the investigation of ethmoidal haematoma, investigation of idiopathic Horner’s syndrome, diagnosis and treatment of vascular anomalies and aneurysms, tumoral embolisation and thrombectomy, have also been described [4,5] in horses. Traditionally, these procedures were carried out by dissection and open arteriotomy of the CCA in the junction of the proximal and middle third of the neck. In these approaches, a 10-cm skin incision is made to perform the dissection of the surrounding muscles and the vagosympathetic trunk. Then the CCA is elevated from the incision to allow the insertion of an introducer sheath into the artery [6,7,8,9,10,11]. After the procedure, the arterial wall must be sutured, as well as the muscular planes and the skin.

Percutaneous ultrasound-guided arterial access has been used in humans [12,13,14,15,16,17], dogs [18], and experimental pigs [19]. In horses, there is a report of a single case from decades ago [20]. Ultrasound-guided percutaneous access has not been mentioned in the literature from that first report [20] in 1998, to 2015, when Maninchedda et al. [21] accurately described the common carotid access for internal carotid coil deployment in anesthetised and standing horses. This procedure avoids the disadvantages of invasive access to the CCA using large skin incisions [21], and the ultrasonography can also facilitate the ability to identify unusual branching at the origin of the internal carotid artery [22]. However, some complications have been described after PUGCA in horses, mainly those related to the possibility of bleeding and haematoma formation [21,23]. These complications can be serious in humans and small animals due to anticoagulation and a lack of collaboration of the patient. In these circumstances, control measures, including open surgery for arterial ligation, may be necessary, increasing morbidity, mortality, and health care costs [24]. In two experimental studies in horses, the rate of haematoma formation in PUGCA seems to be larger than in human patients [21,23].

Since the 1990s, different types of commercial vascular closure devices have been developed for their use in humans [25]. There are several types of vascular closure with different action mechanisms. The best known and often used are (a) surgical stapling, external arterial closure with a circumferential metal clip (e.g., StarClose vascular closure system, Abbott Vascular, Santa Clara, CA, USA), and (b) automated suture systems, with a single polypropylene monofilament (e.g., Perclose Proglide Suture Mediated Closure –SMC-, Abbott Vascular, Santa Clara, CA, USA) and (c) compression of the puncture site in a ‘sandwich technique’ in combination with induction of homeostasis by collagenous sponge (Angio-Seal VIP. Terumo, Leuven, Belgium) [26]. All of them have demonstrated their effectiveness in closing vascular accesses of five to eight French (1.6–2.6 mm). The use of these devices could prevent the potential complications associated with PUGCA in horses.

The objectives of this study are to describe our experience with PUGCA in clinical cases retrospectively and also, for the first time, to report the use, effectiveness and safety of a commercial vascular closure device after percutaneous arterial access in horses.

## 2. Materials and Methods

A retrospective study of clinical cases was performed, evaluating clinical records to determine the feasibility of ultrasound-guided percutaneous access to the CCA and then the safety and complications of the use of a commercial device for arterial puncture closure.

### 2.1. Case Selection

The medical records of horses undergoing arterial access for interventional radiological procedures between 2005 and 2015 at the Equine Surgery and Medicine Service of the Veterinary Hospital of the University of Zaragoza (HVUZ) were retrospectively reviewed.

The inclusion criteria were arterial access performed percutaneously under ultrasound guidance and arterial puncture site closed with the Angio-Seal vascular closure device.

### 2.2. Percutaneous Ultrasound-Guided Carotid Access

Before the intervention, penicillin G sodium (25,000 IU/Kg), gentamicin (6.6 mg/Kg) and flunixin meglumine (1.1 mg/Kg) were intravenously administered.

In all cases, the percutaneous arterial access was performed in the CCA with the horse in lateral recumbency (intervention side on the upper side) under general inhalatory anaesthesia. The head and neck and also the scapular area in one case (haemangiosarcoma in a foal) were supported on a radiolucent table. The anterior third of the neck was aseptically prepared, and the skin was incised. A 0.5 cm skin incision was performed dorsal to the jugular vein between the proximal and medial third of the neck to avoid skin resistance during needle penetration and facilitate the puncture. A linear ultrasound probe was used to identify CCA in a transversal view (Figure 1 and Figure 2). The ultrasound probe pressure was low to avoid modifications of the structures when the probe was going to be retired after the puncture. Therefore, the CCA was punctured using an 18 G needle with ultrasound guidance in a short-axis approach. Once the CCA was accessed, a 0.035” guidewire was advanced into the artery, and the needle was exchanged for a 7 Fr introducer sheath (Performer. Cook Medical. Limerick, Ireland) using the Seldinger technique. Under fluoroscopic guidance, the intervention site was catheterised by means of a 0.035” hydrophilic guidewire (Radiophocus Glidewire. Terumo. Leuven, Belgium) and a 5 F and 65 cm C2 or Vanschie1 angiographic catheters (Beacon Tip. Cook Medical. Limerick, Ireland).

### 2.3. Arterial Puncture Closure

After the interventional or diagnostic procedure, PUGCA was closed using a commercial closure device (Angio-Seal VIP, Terumo. Leuven, Belgium) (Figure 3).

It is composed of an absorbable collagen sponge and a polymer anchor that are connected by an absorbable self-tightening suture. The device seals the arterial hole between the anchor and the collagen sponge by pulling up the anchor from the artery lumen and tamping the sponge on the outside. Angio-Seal VIP insertion requires a specific insertion sheath, the Angio-Seal device and a 0.035” guidewire (Figure 3). The system guidewire was inserted through the 7 Fr sheath used during the endovascular procedure, and then that sheath was removed while compressing the access to avoid bleeding. The Angio-Seal insertion sheath and dilator were advanced over the guidewire into the artery until blood flowed through the distal ending of the Angio-Seal system to check the correct position of the insertion sheath into the vessel. Then, all the assembly was pulled back until the blood flow stopped, indicating that now the distal inner hole was out of the artery, but the dilator was still in the vessel. After that, the assembly was pushed back in until the blood flowed again, verifying this way the accurate position for deployment, with 1.5 cm of the sheath into the artery. With the sheath properly positioned, the guidewire and dilator were removed, and the Angio-Seal device was inserted into the insertion sheath until a ‘click’ was felt.

The anchor should be now correctly positioned into the artery whilst the collagen sponge is still in the sheath. The device cap is then pulled back to position the anchor at a perpendicular angle to the sheath. Then all the systems can be withdrawn until a colour tube appears. This tube is then pushed downwards, maintaining tension with the suture to tamp the collagen sponge on the outer part of the arterial wall. When a black mark appeared in the suture, the tube was removed, and the suture was cut. The main steps of the procedure are illustrated in Figure 4.

### 2.4. Animal Data and Recorded Parameters

Each horse’s data were manually compiled from the clinical files into a database using Microsoft Access. The following demographic data were collected: age, weight, sex (stallion, mare, or gelding) and breed.

The reason for the endoarterial procedure was identified. It was also noted on which side the arterial access was performed and whether it was unilateral or bilateral. In addition, it was recorded whether PUGCA was successful in order to carry out the procedure, as well as variables related to the safety and efficacy of the arterial puncture closure with Angio-Seal: immediate bleeding absence; proper placement by visual inspection and ultrasound examination; no bleeding or swelling in the area after anaesthesia recovery and in mid-term, during hospitalisation, rectal temperature, ultrasonography of the area and local signs.

## 3. Results

A total of 11 cases and 12 PUGCAs (one case with bilateral arterial access) closed with an Angio-Seal VIP device were identified. These procedures were carried out in a mixed population of 11 owned horses with ages ranging from 1 month to 17 years (mean 11.8 years), weighing 100 to 502 kg (mean 424.6 kg), four females and seven males (two stallions and five geldings). The breeds included were four Thoroughbred, one Spanish Pure Breed, one Anglo-Arabian and five mixed breeds. The indications for arterial access were seven guttural pouch artery embolisations (bilateral in one case), three diagnostic arteriographies (two ethmoidal haematomas and one horse with unexplained Horner’s syndrome) and two presurgical tumour embolisations (one foal with a congenital haemangiosarcoma and a mare with mandibular osteosarcoma). In the bilateral case, both accesses were performed 48 h apart, starting on the right side (Table 1).

Accurate ultrasound-guided catheterisation of the CCA was carried out in all cases.

In all cases, 7 Fr introducer sheaths (Performer. Cook Medical. Limerick, Ireland) and 8 Fr Angio-Seal VIP (Terumo. Leuven, Belgium) were used as a closure system (8 Fr Angio-Seal is indicated for closure of 6 to 8 Fr accesses).

In the short term, immediately after catheter removal, in 10 of the 12 PUGCAs (83.33%), no signs of haemorrhage were observed, and ultrasound examinations were normal (Figure 5), with no findings of haematomas.

Bleeding was detected in two arterial access points (2 of 12, 16.66%). In one of them, the bleeding was controlled by manual compression for 10 min, and posterior ultrasound exams revealed a discrete perivascular haematoma and the Angio-Seal anchor outside of the vessel lumen (Figure 6). This was probably due to the excessive traction of the system by the operator. In the other one, bleeding could not be controlled by manual compression. Ultrasound assessment during compression showed that the jugular vein was inadvertently pierced when performing the PUGCA, and the introducer sheath passed through the jugular vein on its way to the carotid artery (Figure 7). This complication occurred at the time of vascular access, but no problems were detected during the procedure. However, at the end of the intervention, the Angio-Seal could not be deployed correctly because the anchor was into the lumen of the CCA, but the jugular vein was between the CCA and the collagen sponge. The device must be removed, and the CCA dissected and ligated due to massive bleeding. Therefore, only 10 of 11 cases remained for anaesthetic recovery and mid-term medical follow-up. None of the animals showed bleeding after anaesthetic recovery nor signs of local swelling, except in the case where manual compression was necessary. In this case, no bleeding was detected, but the area was slightly swollen.

Three horses were euthanatised on the first days after the procedure due to untreatable mandibular osteosarcoma or severe persistent neurological problems after guttural pouch arteries embolisation (in one of them, the guttural pouch mycosis was bilateral). Therefore, mid-term medical follow-up was only possible in 7 of 11 horses (7 of the 12 PUGCA).

During hospitalisation, none of the surgical sites showed any alterations on visual inspection. During this period, ultrasound and specific local checks were carried out on only five of these seven animals. In all these five cases, the ultrasonographic examination showed CCA patency. In the case of haemorrhage controlled by manual compression, the ultrasound showed that the perivascular haematoma had an adequate evolution in this period. In one case, rectal temperature was elevated during the 10 days of medical follow-up after presurgical embolisation of periscapular haemangiosarcoma.

## 4. Discussion

Our results confirm that ultrasound-guided-percutaneous access to the CCA is possible and feasible. In addition, the use of the Angio-Seal arterial puncture closure system, which could help to reduce the potential puncture site complications, is reported for the first time in the horse.

The possibility of reaching this artery by PUGCA without the need for incision and arteriotomy has been previously demonstrated [21] and used [23] on experimental horses, but to our knowledge, this is the first time that its use has been described in a clinical case series. The access was successfully performed in all (12) procedures. Horse’s hard skin can make puncture and sheath introduction difficult, so a short skin incision was performed before puncture to prevent entry difficulties. Excessive tissue compression with the ultrasound probe when puncturing the artery was avoided since it can cause the needle to exit the artery when relieving pressure.

In contrast to other previously described studies [21], the PUGCA was performed with the probe transverse to the direction of the artery. The main disadvantage of the short-axis approach is that there is a part of the needle puncture that is not possible to follow if the surgeon does not have enough training. However, the exact point on the perimeter of the artery where the puncture is performed can be controlled, and it is easier to maintain the same ultrasound plane without overlapping the jugular vein. This axial ultrasound section is the standard one used in human medicine for femoral artery puncturing [24].

In all procedures, angiography could be performed without complications. The introducer sheath, guidewire and interventional catheter could be placed and used without difficulty. All endovascular manoeuvres were possible, even in the case of jugular piercing, because the use of the introducer sheath maintained the arterial and venous access perfectly sealed until it was removed. At the moment of this complication, surgeons did not have enough training in puncturing the artery at this depth, and it was probably due to using a short-axis ultrasound approach. To avoid it, the ultrasound probe should follow the tip of the needle in all the way along from the skin to the CCA to ensure that no other structures are pierced. Ultrasound monitoring of the arterial access with the probe placed longitudinally could have helped to avoid this complication.

The advantages of this percutaneous ultrasound-guided arterial access have been already described: minimally invasive approach with minimal tissue dissection, which should decrease the time required for tissue healing, the risk of tissue infection, wound dehiscence, or vagosympathetic trunk and recurrent laryngeal nerve damage [21]. However, potential complications, such as the one described in one of our cases, remind us that one of the disadvantages of this procedure is its technical difficulty and the necessary experience and learning curve of the surgeon, as recognised in human medicine for similar interventions [27].

The commercial vascular closure device Angio-Seal was used in all these clinical cases. We decided to use a closure system due to the high complication rates at the femoral puncture site reported in human medicine and considering that the horse carotid was a larger and deeper artery, which may make it more difficult or inefficient than external compression [28]. At the time these procedures were performed, there were no reports of this type of PUGCA in horses, and the possible incidence of puncture site complications in this species was unknown. The main complications described in humans were bleeding, haematoma, vessel occlusion, dissection, and pseudoaneurysm formation [29]. The reported incidence of such complications in human medicine is usually between 1.3 and 3.4% [30,31]. Most of these complications are bleeding and haematomas [32]. Hemostasis at the access site has traditionally been achieved by manual compression, but several studies suggest that these complications can be reduced by using different commercial arterial puncture site closure systems [25,26]. One of the most widely used systems is the Angio-Seal device, which has been shown to be more effective than manual compression in numerous studies with human patients [33] and experimental pigs [34].

Maninchedda et al. [21] described that this type of arterial access in horses had a haematoma complication rate of 33% (4 out of 12). This incidence is much higher than in human studies, in which the highest rates reach 15.5% [26]. In contrast to human medicine, all of these complications in horses occurred at the time of entry and not at the end of the procedure, after the removal of the catheter from the artery [21]. In another study using PUGCA in seven experimental horses, a total of six complications were recorded, of which four were haematomas (incidence 57%) [23]. One of these haematomas occurs at the access site, but the rest are at the end of the procedure. This is in agreement with what has been observed in other species but with a higher incidence.

In our work, after closing the arterial puncture using Angio-Seal, bleeding was detected in two procedures (16.7%), and both were related to technical mistakes due to lack of training. In the first one, the mistake was that the device was excessively pulled back. The bleeding was controlled by manual compression for 10 min. The other one is the mentioned case where the jugular vein was pierced by the introducer. As a consequence, the collagen sponge of the closure device could not compress the arterial wall properly, and CCA had to be dissected and ligated due to massive bleeding. Proximal compression of the artery was performed until the arterial bleeding could be controlled by open surgery. This complication was more complex to solve than if manual compression alone had been used. In the two experimental studies on horses using PUGCA, no special system for puncture site closure was used [21,23]. Nevertheless, the haematomas resolved without complications.

In human medicine, commercial vascular closure devices offer the potential for the improved control of the percutaneous access site haemostasis and reduce complications compared to manual compression. However, while randomised clinical trials have shown reductions in time to haemostasis and ambulation, the data do not demonstrate consistent reductions in arterial puncture site complications or improvements in clinical outcomes [12]. There is no literature on horses comparing the two approaches, nor does our study design attempt to compare the two methods. Nevertheless, when contrasting our findings with those of the two previous articles on horses using PUGCA [21,23], the need for the routine use of Angio-Seal or other devices for percutaneous arterial access closure in this species can be questioned. However, it should be noted that these articles do not specify how long the manual compression was maintained. In human medicine, after femoral arterial accesses, a compression time of approximately 30 min is usually necessary, followed by at least 4 h of immobilisation [25]. In addition, these studies were carried out on healthy experimental horses. It would be possible that manual compression alone is not sufficient in horses with pathologies, such as mycosis of the guttural pouch, where severe haemorrhages could lead to significant haemodynamic and coagulation disturbances [23].

This is related to some of the limitations of our work. It would have been desirable to include in our retrospective study a group of clinical case controls in which PUGCA had been performed but in which Angio-Seal had not been used. It would have been interesting to have coagulation data in these clinical cases.

## 5. Conclusions

Our experience in a clinical case series confirms previous experimental results, which showed that percutaneous ultrasound-guided carotid access is feasible and effective in horses undergoing endovascular procedures, avoiding open arteriotomy.

To seal the arterial puncture site, regardless of possible complications due to incorrect use, the Angio-Seal human medicine device can be used safely and effectively in horses. However, further studies comparing arterial access site management using this device or using manual compression would be necessary to state if its routine use in horses is advisable.

## Figures and Tables

**Figure 1 animals-12-01481-f001:**
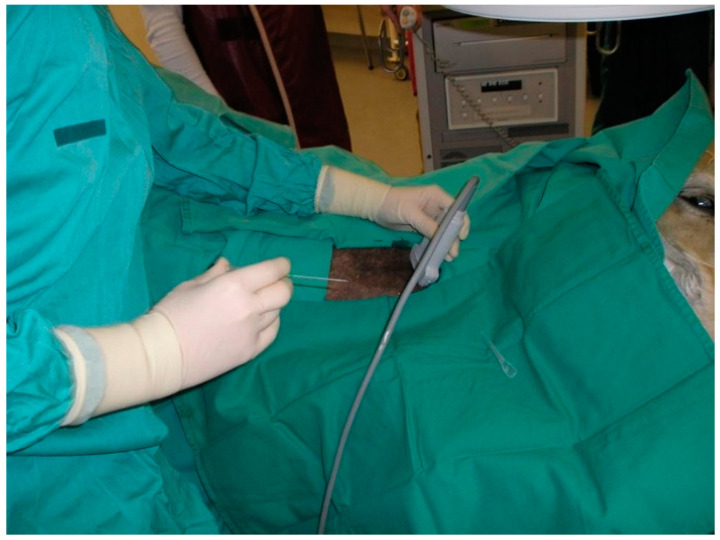
Percutaneous ultrasound-guided puncture to the right common carotid artery, with the horse under general anaesthesia in lateral recumbency. The ultrasound probe was previously immersed in disinfectant solution and rinsed with sterile wash water.

**Figure 2 animals-12-01481-f002:**
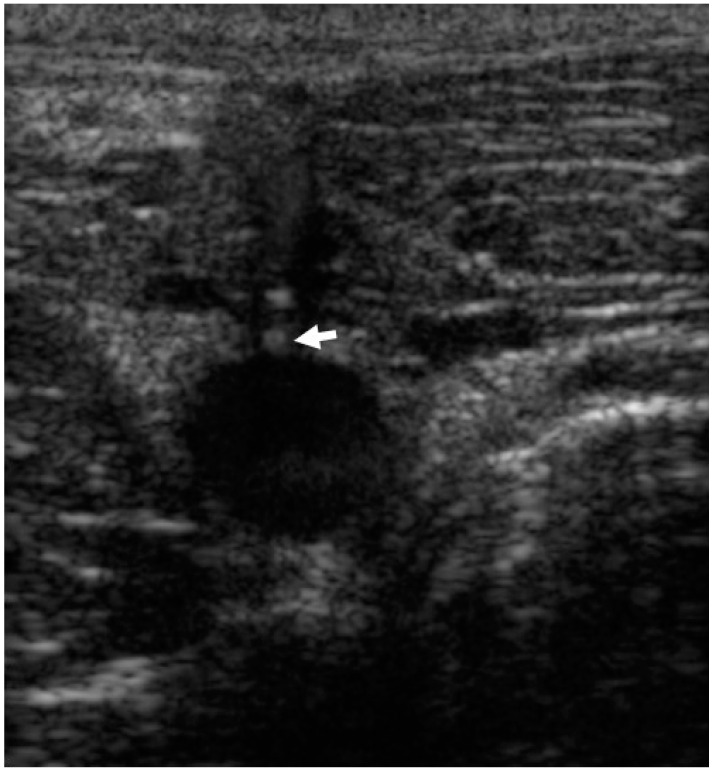
Transverse ultrasonography of the common carotid artery during percutaneous ultrasound-guided access. The tip of the needle is just above the artery (arrow).

**Figure 3 animals-12-01481-f003:**
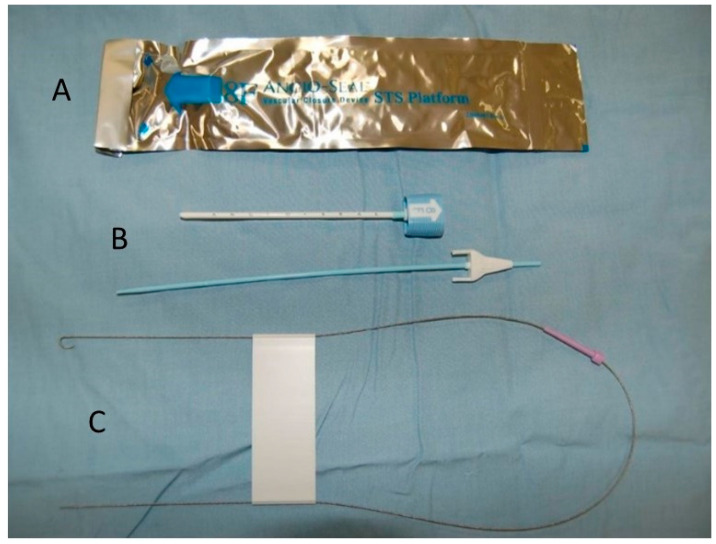
Angio-Seal arterial puncture closure device commercial kit, including: Angio-Seal VIP device (**A**), insertion sheath and dilator (**B**) and 0.035” stainless steel guidewire (**C**).

**Figure 4 animals-12-01481-f004:**
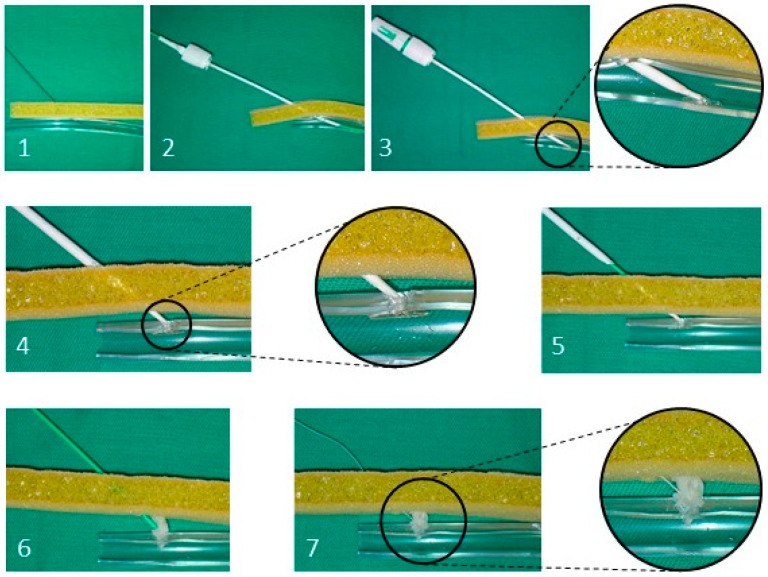
Schematic representations of the main steps in the application of the Angio-Seal VIP system. The yellow cancellous material represents the skin and subcutaneous tissue. The transparent plastic tube represents the artery. (**1**) guidewire advanced into the artery after withdrawal of the former introducer sheath. (**2**) Angio-Seal insertion sheath advanced over the guidewire (at this moment, blood flow through the locator confirms the proper position of the sheath). (**3**) insertion of Angio-Seal device into the sheath and deployment of the anchor. (**4**) Angio-Seal device is pulled back until the anchor is properly located. (**5**) all the assembly is pulled back until the colour tube appeared. (**6**) advance the compaction tube until feeling resistance. (**7**) compaction tube has been removed and collagen sponge has been deployed in place, and it is attached to the suture that still has to be cut.

**Figure 5 animals-12-01481-f005:**
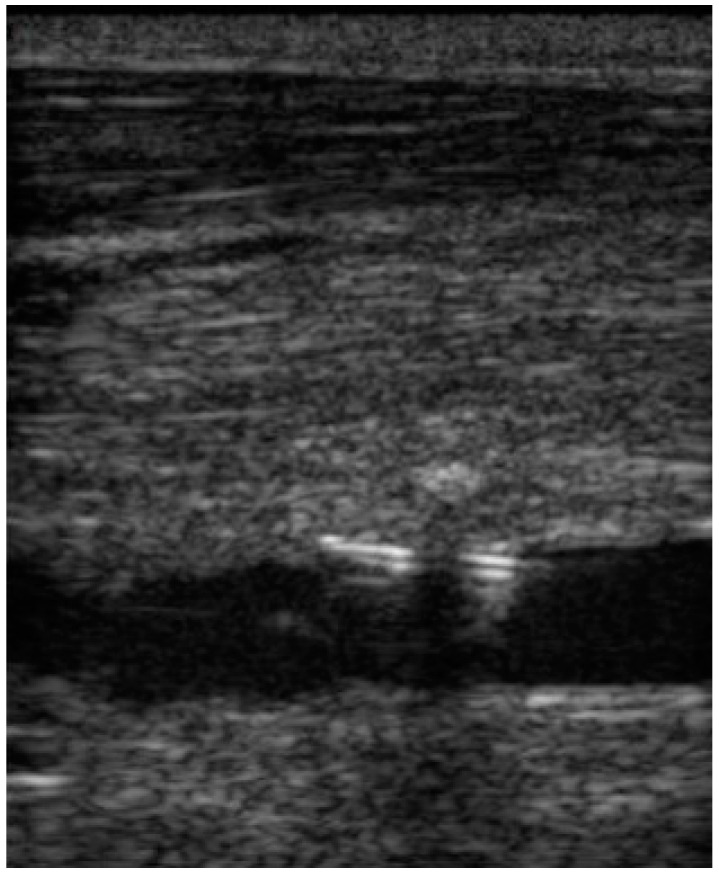
Longitudinal ultrasound image with the Angio-Seal device properly positioned for closure of the common carotid artery puncture.

**Figure 6 animals-12-01481-f006:**
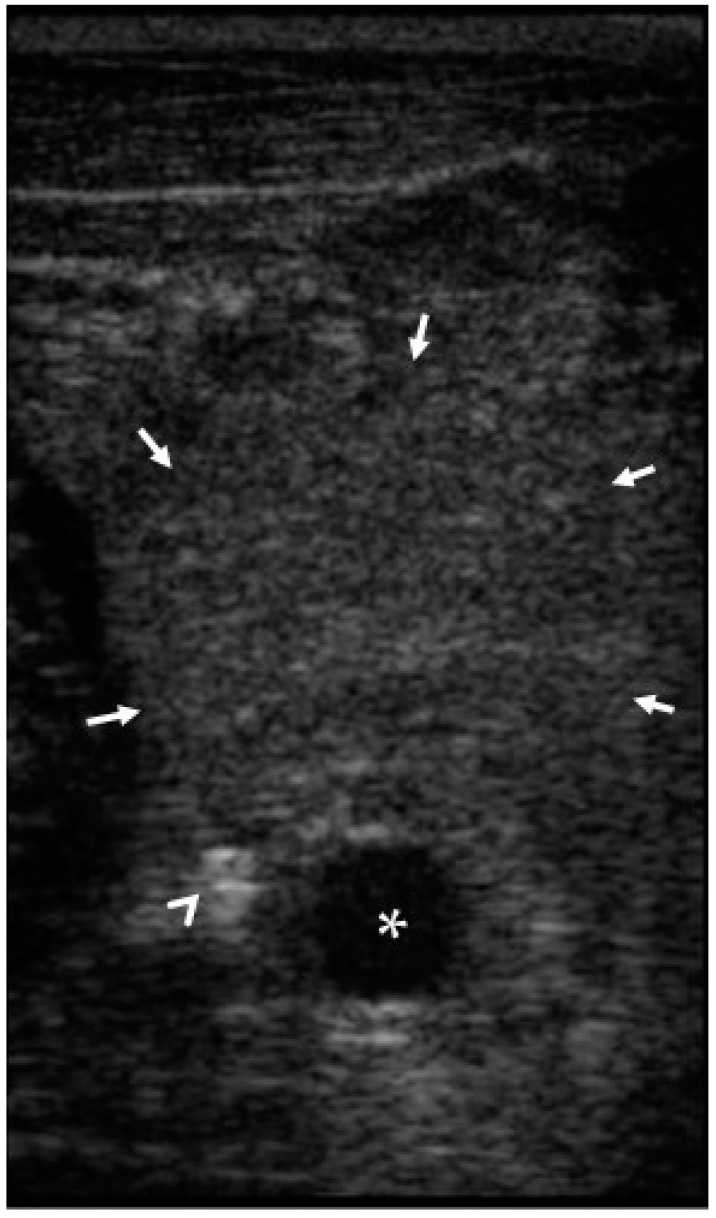
Case 2. Ultrasound showing a perivascular haematoma (delimited by arrows) around common carotid artery (asterisk). Angio-seal anchor (arrowhead) appears to be outside of the arterial lumen.

**Figure 7 animals-12-01481-f007:**
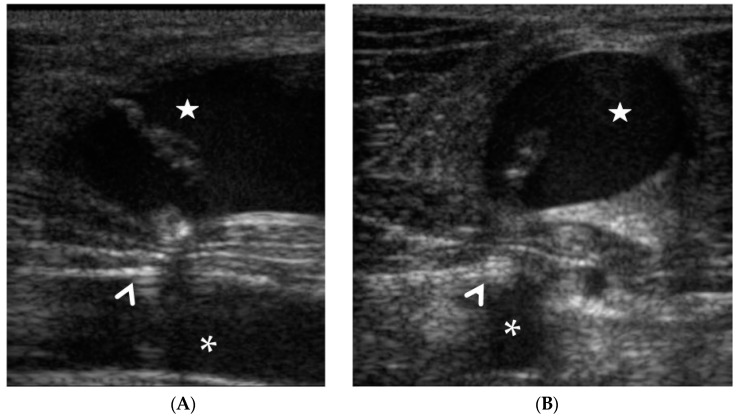
Case 5. Longitudinal (**A**) and transverse (**B**) ultrasound showing incorrect deployment of the Angio-Seal system (arrowhead): the jugular vein (star) was inadvertently pierced by the introducer during percutaneous ultrasound-guided carotid (asterisk) access.

**Table 1 animals-12-01481-t001:** Access side, indication and findings registered after the intervention.

Case	Side	Indication	Immediate	Post-Recovery	During Hospitalisation		
					RT	Local	Ultrasonography
1	Right	GPAE					
2	Left	DA	Bleeding	LS			Haematoma
3	Left	GPAE					
4	Left	PSTE			Fever		
5	Left	GPAE	Bleeding	Without data: AS badly deployed and removed			
6	Right	GPAE					
7	Bilateral	GPAE			Without data: euthanatised		
8	Left	DA					
9	Right	DA					
10	Right	GPAE			Without data: euthanatised		
11	Left	PSTE			Without data: euthanatised		

Empty cells: no abnormal findings. RT: rectal temperature. GPAE: guttural pouch arteries embolisation. DA: diagnostic angiography, PSTE: presurgical tumour embolisation. LS: local swelling. AS: Angio-Seal.

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
