# Peer review of "Percutaneous Ultrasound-Guided Carotid Access and Puncture Closure with Angio-Seal in Horses"

_animals, 2022, doi:10.3390/ani12121481_

Round 1

Reviewer 1 Report

Dear Authors and Editors, thank you for the opportunity to review this interesting work. The ides is very nice, the design is sound and the results are interesting. Although the purpose of the paper is to describe the use of the Angio-Seal device, I think it would be interesting for many to read alongside also a more detailed description of the entire procedures that were carried on, for which vascular access was necessary (lines 119-120 to be expanded). I appreciate that the authors admit that they are not sure whether a sealing device is overall necessary for these size vascular access. Nevertheless I think this is definitely a worthy investigation and publication.   Please spell check, there are minor english mistakes.

1. I believe this is a question for the authors, more so than for the reviewer. I believe the manuscript aims to test the safety and practicality of the illustrated device in equine vascular surgery. 2. I do consider the topic interesting and relevant as vascular surgery does happen in equine surgery and it is very rudimental so far. 3. i believe this question would be also better answered from the authors than from the reviewer. A vascular sealing device has never been described in equine vascular surgery so far. It is unclear whether it is necessary or not, but it is good to have a system that is proven to be safe to use in case of necessity. 4. I appreciate that the study was carried on clinical cases and no lives were sacrificed to the methodology. I feel that this choice is much more ethical than using experimental animals, even though a bit of the randomization is lost. The only improvement I would suggest, as I stated in my first review, is to expand on the vascular procedures that were carried on before using the sealing device. 5. The conclusions are consistent with the evidence. 6. The references are appropriate

Author Response

Thank you for your comments.

Although the purpose of the paper is to describe the use of the Angio-Seal device, I think it would be interesting for many to read alongside also a more detailed description of the entire procedures that were carried on, for which vascular access was necessary.

Even if it is not the purpose of the paper, following your comment, we have added more information on the clinical reasons why vascular access was necessary:

“The indications for arterial access were 7 guttural pouch arteries embolizations (billateraly in one case), 3 diagnostic arteriographies (two ethmoidal haematomas and one horse with unexplained Horner's syndrome) and 2 presurgical tumour embolizations (one foal with a congenital haemangiosarcoma and a mare with a mandibular osteosarcoma).”

Please spell check, there are minor english mistakes.

We have revised the manuscript and corrected spelling mistakes.

Reviewer 2 Report

I think the study is interesting although including a larger number of patients and having data on a control group would have allowed us to have a clearer picture of the validity of the method adopted.

Author Response

I think the study is interesting although including a larger number of patients and having data on a control group would have allowed us to have a clearer picture of the validity of the method adopted.

Thank you for your comments.

We agree that, indeed, having more cases and a control group would add strength to the work. However, the casuistry of this type of endovascular interventions in horses, at least in our environment, is scarce

Reviewer 3 Report

Dear authors

An interesting manuscript.

Couple of suggestions in the pdf, mainly typing errors I presumed.

Author Response

Thank you for your comments.

We have corrected the typographical and spelling errors in lines 48, 72, 129, 148, 149, 208 and 265.

The phrase in lines 63 to 65 has been reworded to read as follows:

“Ultrasound guided percutaneous access has not been mentioned in the literature from that first report [20] in 1998, to 2015, when Maninchedda et al [21] accurately describes the common carotid access for internal carotid coil deployment in anesthetised and standing horses.”

Regarding your comment on the sterility of the ultrasound probe in Figure 1:

The ultrasound probe was previously immersed in disinfectant solution and rinsed with sterile wash water. Please let us know if you consider that this information should be included in the footnote of the figure.